# Mapping quantitative trait loci and predicting candidate genes for leaf angle in maize

**Ning Zhang, Xueqing Huang** *

State Key Laboratory of Genetic Engineering, School of Life Sciences, Fudan University, Shanghai, China

* xueqinghuang@fudan.edu.cn

## Abstract

Leaf angle of maize is a fundamental determinant of plant architecture and an important trait influencing photosynthetic efficiency and crop yields. To broaden our understanding of the genetic mechanisms of leaf angle formation, we constructed a $F_{3:4}$ recombinant inbred lines (RIL) population to map QTL for leaf angle. The RIL was derived from a cross between a model inbred line (B73) with expanded leaf architecture and an elite inbred line (Zheng58) with compact leaf architecture. A sum of eight QTL were detected on chromosome 1, 2, 3, 4 and 8. Single QTL explained 4.3 to 14.2% of the leaf angle variance. Additionally, some important QTL were confirmed through a heterogeneous inbred family (HIF) approach. Furthermore, twenty-four candidate genes for leaf angle were predicted through whole-genome re-sequencing and expression analysis in qLA02-01and qLA08-01 regions. These results will be helpful to elucidate the genetic mechanism of leaf angle formation in maize and benefit to clone the favorable allele for leaf angle. Besides, this will be helpful to develop the novel maize varieties with ideal plant architecture through marker-assisted selection.

## Introduction

Maize (Zea mays L.) is one of the most important cereal crops worldwide, and increasing the grain yield has been the most important goals of maize production [1]. Among the various traits that are normally considered in maize breeding programs, the leaf angle (LA), defined as the angle of leaf bending away from the main stem, is an important trait influencing plant architecture and yield production [2,3]. The less of leaf angle is, the more upright the leaves are. Upright leaves can maximize photosynthesis efficiency through maintaining light capture and reducing shading as canopies went more crowded, which in turn increase yield production in high density cultivation [3–6]. Therefore, an appropriate leaf angle is a prerequisite for attaining the desired grain yield in maize-breeding projects. A more thorough understanding of the molecular and genetic mechanism determining leaf angle will contribute to develop novel maize varieties with ideal plant architecture.

Genetic studies have indicated that leaf angle in maize is a complex trait controlled by both qualitative genes and quantitative genes. According to an incomplete statistic, eleven representative maize genes that control the leaf angle have been cloned, five of which were identified by mutagenesis: *knox* [7], *liguleless1* (*lg1*) [8], *liguleless2* (*lg2*) [9], *liguleless3* (*lg3*) [10], *liguleless narrow* (*lgn*) [11], and six were resolved through QTL-cloning approach: *ZmTAC1* [12],

**Data Availability Statement:** All relevant data are within the paper and its Supporting Information files.

**Funding:** This work was funded by National Natural Science Foundation of China (grant number

31471151). The funder had no role in study design, data collection and analysis, decision to publish, or preparation of the manuscript.

**Competing interests:** The authors have declared that no competing interests exist.

*ZmCLA4* [13], *ZmILI1* [14], and *UPA2/UPA1* [15], *ZmIBH1-1* [16]. In the past 30 years, a large number of QTL for leaf angle have been obtained by genetic dissection of maize leaf angle using biparental populations [17–24]. Mickelson et al. firstly identified nine leaf angle QTL which were distributed on six chromosomes in two environments using the RFLP marker technique in the B73 × Mo17 population containing 180 RILs [17]. Utilizing $1.49×10^6$ single nucleotide polymorphism (SNP) markers, Lu et al. identified 22 SNP that were significantly associated with leaf angle and located on eight chromosomes, explaining 21.62% of the phenotypic variation [25]. The natural variations in leaf architecture were also discovered in connected RIL populations in maize. Tian et al. used nested association mapping (NAM) population from 25 crosses between diverse inbred lines and B73 to conduct joint linkage mapping for the leaf architecture, and identified thirty small-effect QTL for leaf angle [26]. A total of 14 leaf angle QTL were also identified using a four-way cross mapping population [27]. The large numbers of QTL for leaf angle detected in various mapping populations strengthen the understanding of the genetic mechanism of leaf angle in maize. However, different results were provided by different studies, including QTL number, location, and genetic effect. Inconsistent results of QTL detection in different study shown the importance and necessity of QTL mapping to uncover the tangled genetic mechanism of leaf angle. Therefore, taking the polygenic and complex inheritance nature of maize's leaf angle into consideration, further investigating the QTL that underlie the trait's phenotypic variance is required.

In the present study, a $F_{3:4}$ RIL population derived from a cross between inbred line B73 and Zheng58, was constructed to identify QTL for leaf angle. The objective of the study is to further elucidate the genetic architecture that underlie leaf angle, and to further evaluate and confirm the genetic effect of QTL allele through heterogeneous inbred family approach. It is expected that the further study into the genetic mechanism that underlies the leaf angle could provide candidate genes for maize breeding projects.

## Materials and methods

### Plant materials

The recombination inbred line population was constructed by crossing Zheng58 with B73. The two parents were selected on the basis of maize germplasm groups and their different leaf architecture. Zheng58 is an elite foundation inbred line with compact leaf architecture, as female parent of a famous maize variety Zhengdan958 in China. B73 is a model inbred line with expanded leaf architecture and has been sequenced [28]. A single seed descent from one $F_1$ progeny and then two generations of self-pollination were applied to produce the recombination inbred line population with 165 lines [29].

### Field experiments and statistical analyses

The trials were conducted at the Songjiang experimental station in Shanghai (121˚45′E, 31˚ 12′N) from April to September during 2014 and 2015, where experimental field bases have been set up by the school of life sciences, Fudan University. More than two hundred lines were planted and 165 lines were survival. The school of life sciences was approved for field experiments, and the field studies did not involve protected or endangered species. The randomized complete block design with two replications was employed. Every plot had a row of three meters long and 0.67 meters wide with a planting density of 50,000 plants per hectare. Corn field management was in accordance with traditional Chinese agricultural production management methods.

Ten days after pollination (DAP), three plant representatives from the middle of each plot were randomly selected to evaluate the leaf angle. We measured three leaves and used the

mean value for data analysis. Traditionally, leaf angle was assessed by measuring the angle of each leaf from a plane defined by the stalk below the node subtending the leaf. Three consecutive leaves were measured for each plant, including the first leaf above the primary ear, the primary ear leaf and the first leaf below the primary ear. The leaf angle data for each of RIL populations was averaged for the three measured plants.

To further verify the authenticity of the QTL mapping results and the allelic effects of the leaf angle QTL, we constructed six heterogeneous inbred family (HIF) lines [30] from $F_3$ RIL segregating for target leaf angle locus but homozygous for the other major leaf angle loci. Each heterogeneous inbred family consisted of at least 120 plants and each individual from the progeny of these heterogeneous inbred family was genotyped using a molecular marker. And the molecular markers were tightly linked to the target QTL. Meanwhile, the leaf angle phenotype as described above was measured. ANOVA was used to compare and analyze phenotypic differences between different homozygous lines isolated in the target QTL region.

Statistical analysis of the phenotypic mean data measured in the population was performed using SPSS 20.0 software. The broad-sense heritability ($h^2$) was estimated as the proportion of variance explained by between RIL (genotypic) variance and RIL by block (error) variance.

## Genetic map construction and QTL mapping

Samples for DNA extraction were collected at the four-leaf stage of the seedlings in the RIL population and parent line, and genomic DNA was isolated using the CTAB method [31]. The $F_3$ plants were genotyped using SSR markers. Primer sequences are available from the Maize Genetics and Genomic Database (Maize GDB).

The software package MapQTL6.0 was used to identify and locate QTL on the linkage map by using interval mapping and multiple-QTL model (MQM) mapping methods as described by Churchill et al. [32] and Huang et al. [33]. LOD threshold values applied to declare the presence of QTL were estimated by performing permutation tests implemented in Map QTL 6.0 using at least 1000 permutations of the original data set, resulting in a 95% log 10 of the odds ratio threshold values of 2.9. Using MQM mapping, the percentage of variance explained and the estimated additive genetic effect by each QTL and the total variance explained by all the QTL affecting a trait were obtained [33].

## DNA library construction and Whole-genome re-sequencing analysis of Zheng 58

Extraction of total genomic DNA from young leaves of inbred line Zheng 58 by modified CTAB method [31]. Separation of genomic DNA used to generate sequencing libraries. The constructed library was first subjected to library quality checks, and the quality qualified library was sequenced by HiSeq system using standard protocols.

The raw readings (double-ended sequences) obtained by sequencing were quality assessed and filtered to obtain Clean Reads for subsequent bioinformatics analysis. The Burrows-Wheeler Alignment (BWA) software aligns the short sequences obtained from the second-generation high-throughput sequencing with the reference genome. The Clean Reads were compared with the reference genome sequence. The SNP and Small InDel were detected and annotated according to the comparison results.

## Total RNA extraction and expression analysis of candidate genes

Fresh leaves in V5 stage were sampled from the B73 and Zheng58 [15]. According to the manufacturer's instructions, total RNA was isolated from sampled fresh leaves using Fast Pure Plant Total RNA Isolation Kit (Vazyme, RC401). 1.0 μg of each sample total RNA was reverse

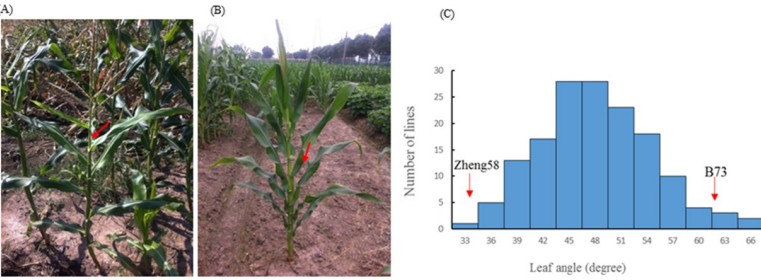

**Fig 1. The maize leaf angle trait.** (A) Leaf angle phenotype of the expanded inbred line B73. (B) Leaf angle phenotype of the compact inbred line Zheng58. (C) The frequency distribution of leaf angle within the $F_{3:4}$ population. The red arrow refers to the mean value of the leaf angle phenotype of the compact inbred line Zheng 58 and the expanded inbred line B73.

transcribed (Vazyme, R323), and then performed Quantitative PCR (Vazyme, Q711). 18S was selected as housekeeping genes for the internal control, and the primers used in the RT-qPCR are listed in S1 Table. Three independent biological replicates were collected for each sample. The candidate genes expression levels were quantified with the comparative CT($2^{-\triangle\triangle CT}$) method.

## Results and discussions

### Analysis of leaf angle in $F_{3:4}$ population and parental lines

There was significant difference in leaf angle between the two parents B73 and Zheng58. Zheng58 had compact leaf architecture (Fig 1A) with an average leaf angle of 31˚, whereas B73 displayed expanded leaf architecture (Fig 1B) with an average leaf angle of 62˚ (Table 1). Table 1 presented the descriptive statistics of leaf angle for the two parents and the $F_{3:4}$ population. The wider range of variation for leaf angle in the $F_{3:4}$ population was observed, and normal distribution with transgressive segregation suggested polygenic inheritance of the trait (Fig 1C). The calculated broad-sense heritability ($h^2$) value for leaf angle trait was high as shown in Table 1.

### Detection of the leaf angle QTL

189 SSR markers with polymorphic between the two parents were identified by the screen of 393 SSR primer pairs which evenly distributed on the genome of maize genome. These markers were assigned to corresponding chromosome based on their physical position. The total length of the physical map was 2,058.59 Mb. The number of molecular markers distributed on each chromosome varied from 13 to 29, with an average of 18.8. The average physical distance between two adjacent markers was 10.79 Mb, with the shortest marker interval of 1.42 Mb and the longest 38.78 Mb, and the distribution of all markers in chromosome was not crowded and relatively evenly distributed (Fig 2).

**Table 1. Descriptive statistical analysis of phenotypic values of leaf angle in parents and $F_{3:4}$ population.**

| Trait | B73[a] | Zheng58[a] | $F_{3:4}$ population | | | | | | |
|---|---|---|---|---|---|---|---|---|---|
| | | | Max | Min | Mean | SD | Kurtosis | Skewness | $h^2$ (%) |
| [b]LA | 62.78 | 31.23 | 76.00 | 32.00 | 49.81 | 6.81 | 0.87 | 0.42 | 80.20 |

[a]The data corresponding to leaf angle of the two parents are average values; $P < 0.01$.

[b]LA: leaf angle.

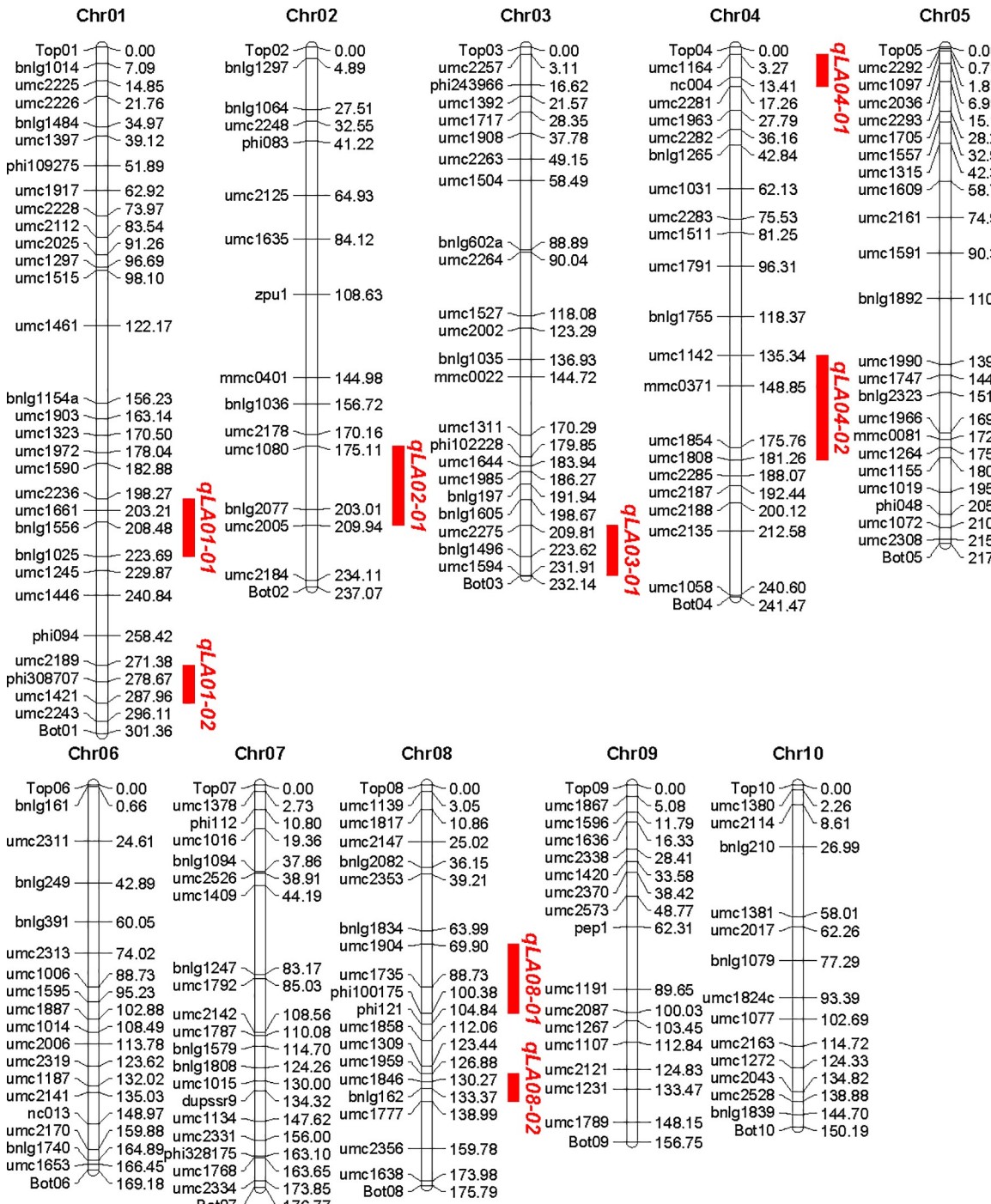

**Fig 2. Construction of genetic linkage maps and mapping of QTL for controlling leaf angle in Maize.**

QTL analysis of leaf angle was conducted using Map QTL 6.0 software. Eight QTL for leaf angle on chromosome 1, 1, 2, 3, 4, 4, 8 and 8 were detected (Fig 2), respectively, which explained 65.4% of the total phenotypic variance, and each QTL explained phenotypic variance ranging from 4.3 to 14.2% (Table 2). It was noteworthy that all QTL had positive additive effects, suggesting that the B73 parent contributed most alleles for increasing leaf angle (Table 2).

**Table 2. Analysis of QTL for controlling leaf angle.**

| QTL | Chr. | Position (Mb) | Marker interval | The nearest markers to QTL | LOD | Additive effect | Explained variance % |
|---|---|---|---|---|---|---|---|
| qLA01-01 | 1 | 211.78 | umc2236-bnlg1025 | bnlg1556 | 4.73 | 3.253 | 8.8 |
| qLA01-02 | 1 | 285.67 | umc2189-umc2243 | umc1421 | 2.96 | 2.677 | 4.3 |
| qLA02-01 | 2 | 195.48 | umc1080-umc2005 | bnlg2077 | 7.55 | 4.109 | 14.2 |
| qLA03-01 | 3 | 220.49 | umc2275-umc1594 | bnlg1496 | 3.16 | 2.803 | 5.8 |
| qLA04-01 | 4 | 10.68 | umc1164-umc2281 | nc004 | 4.95 | 3.445 | 9.1 |
| qLA04-02 | 4 | 156.88 | umc1142-umc1808 | umc0371 | 3.67 | 3.082 | 6.7 |
| qLA08-01 | 8 | 85.88 | umc1904-phi100175 | umc1735 | 5.2 | 3.687 | 10.3 |
| qLA08-02 | 8 | 132.72 | umc1959-umc1777 | bnlg162 | 3.24 | 2.946 | 6.2 |

Additive effect: A positive value indicates that the B73 allele increases the value of the trait; A negative value indicates that the Zheng58 allele increases the value of the trait.

Compared to previous studies (Fig 3), five of the eight QTL for leaf angle were found to have similar chromosomal locations with different mapping experiments or different genetic background. The result demonstrated that the chromosome regions for these consistent QTL might be hot spots for the important QTL for leaf angle. Also the congruence in QTL detected in this study with previous reports indicated the robustness of our results. However, in our study, no QTL was detected on chromosome 5, 6, 7, 9 and 10, which may be due to the too small genetic effects or no allelic difference between the two parents. Interestingly, three QTL were detected on bottom of chromosome 3 (qLA03-01), which shared an interval of 22.10 Mb from 209.81 to 231.91 Mb, middle of chromosome 4 (qLA04-02), which shared an interval of 45.92 Mb from 135.34 to 181.26 Mb, and middle of chromosome 8 (qLA08-01), which shared an interval of 30.48 Mb from 69.90 to 100.38 Mb, respectively. These three QTL have not been reported in previous researchers (Figs 2 and 3). These novel QTL might be due to the specific genetic background from parent Zheng58 with compact leaf architecture. Furthermore, newly detected major QTL may serve a complementary role in revealing the genetic mechanism of leaf angle trait.

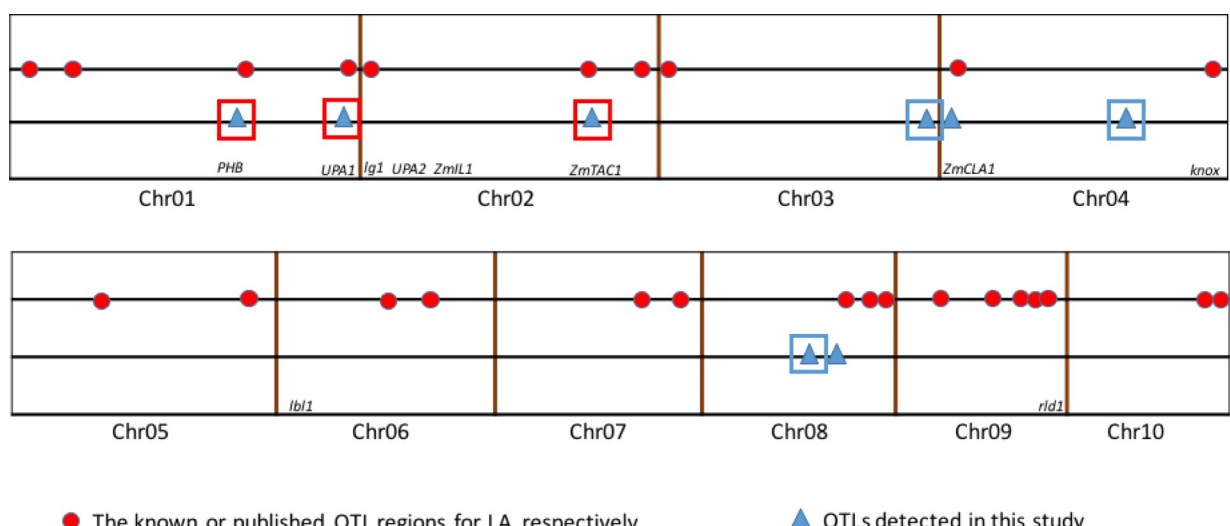

**Fig 3. Comparison of QTL mapping results in this study with the results reported by previous researchers.**

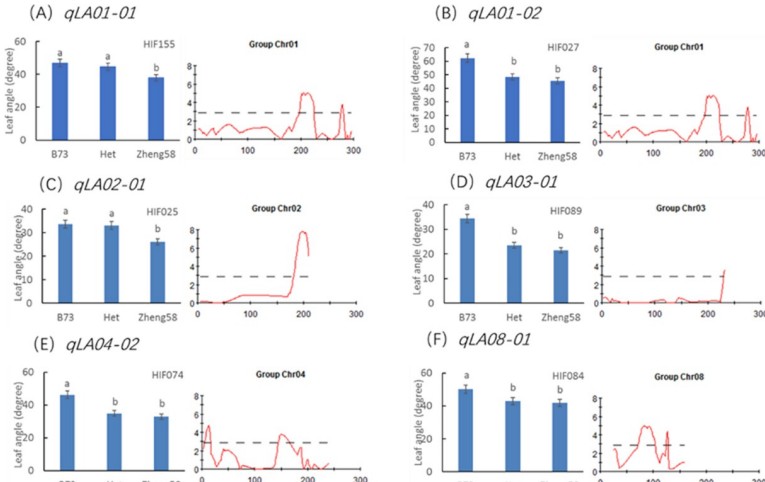

**Fig 4. Significant analysis of the mean values of different genotypes in the same HIF population.** The same letter indicates no significant difference between them, and different letters indicate a significant difference (P value < 0.05).

**Confirmation of QTL for leaf angle.**   There were some heterozygous regions in the genome of the $F_4$ plants. These regions can be applied to validation of QTL through a HIF strategy as described by Tuinstra et al. in 1997 [30]. The HIF strategy has been widely practiced in QTL confirmation and fine mapping [34,35]. In this study, we tried to apply the HIF strategy to validate the allelic effects of six major QTL for leaf angle. HIF segregating for the target QTL to be validated but homozygous for other major QTL regions were chosen. For instance, a RIL (HIF155) with heterozygous region in qLA01-01 was chosen to develop HIF. Theoretically, the leaf angle trait in progenies from selfed HIF155 will be only segregated for qLA01-01. We developed the other five HIFs in the same way. HIF027, HIF025, HIF089, HIF074 and HIF084 were heterozygous at marker umc1421 (qLA01-02), bnlg2077 (qLA02-01), bnlg1496 (qLA03-01), umc0371 (qLA04-02) and umc2147 (qLA08-01), respectively. The overview result for all HIF was presented in Fig 4. In the progenies of all HIFs, the plants carrying B73 alleles at the respective target QTL displayed a significantly bigger leaf angle than those carrying Zheng58 alleles. Therefore, the allelic effects of the major QTL were preliminarily validated.

Considering that the phenotype of leaf angle was easily affected by environmental conditions, we planted these 165 lines in the same season and the same field in 2014 and 2015. These 165 lines showed good repeatability in the phenotype of leaf angle and the phenotypic data were used to detected the QTL for leaf angle. The environmental factors affecting leaf angle phenotype were minimized and the QTL X environment effect was not detected. Subsequently, we construct the HIF populations to verify the major QTL, each HIF population with at least 120 plants. These plants were also grown in the same season and the same field in 2016. The results verified the authenticity of the QTL mapping results and showed that these major QTL are stable and inheritable. In this work, the character of the leaf angle trait resulted from the genetic characteristics of selected parents, Zheng58 and B73.

**Whole-genome re-sequencing.**   In order to compare the genomic sequence differences between the two parents and predicted candidate genes for target QTL, Zheng58 was conducted the whole-genome re-sequencing. After filtering, 91.69 G bp Clean-Base was obtained for subsequent data analysis and the Q30 ratio reached 92.93%, with the 91.68% (at least one base coverage) genome coverage (S2 Table). By aligning against the B73 reference genome, the genome coverage on average for the reference up to 98.83%, which can give 37× average

genome sequencing coverage depth (S1 and S2 Figs). It can be seen from the figure (S2 Fig) that the genome is covered more uniformly, indicating that the sequencing randomness is better. The uneven depth on the map may be due to repeated sequences and PCR preferences. These data demonstrated that the whole-genome re-sequencing data was robust and can be used to subsequent candidate genes analysis.

**Candidate genes prediction in major QTL qLA02-01 region.**   Comparing the genomic sequence differences in the qLA02-01 region between the two parents, we found that 156 genes were variated in the coding region, of which 18 stop gained SNP and 8 start lost SNP were related to 30 genes and 254 InDel (including 6 stop gained InDel, 2 start lost InDel and 246 frame-shift InDel) were related to 126 genes. Absolutely, stop gained and start lost, as well as frame-shift mutations often have a greater impact on causing changes in gene function. Thence, these mutations stimulated our further research interest. Although the mechanism underlying leaf angle is still unclear, previous studies have shown that genes associated with cell cycle, cell size, gravity and plant hormones may participate in regulation of leaf angle development [36]. By analysis of biological processes in the GO annotation clustering results, combined KEGG metabolic pathway analysis, we screened 33 GO enriched genes in the major qLA02-01 region (Table 3). With the help of GO term and functional annotation of candidate genes, ultimately six candidate genes are targeted: *Zm00001d005803*, auxin-activated signaling pathway (GO:0009734); *Zm00001d005888*, response to abscisic acid (GO:0009737); *Zm00001d005889*, oxidation-reduction process (GO:0055114); *Zm00001d006274*, auxin-activated signaling pathway (GO:0009734); *Zm00001d006494*, response to karrikin (GO:0080167); *Zm00001d006587*, response to blue light (GO:0009637).

In addition, variations in gene expression levels may also cause changes in gene function. Therefore, we also screened for genes with mutations in the promoter region, and 21 genes showed GO enrichment (Table 4). By analyzing the gene expression level of genes with mutations in the promoter region, we identified seven genes with significantly different expression levels (Fig 5): *Zm00001d005803*, auxin-activated signaling pathway (GO:0009734); *Zm00001d005818*, response to desiccation (GO:0009269); *Zm00001d005823*, oxidation-reduction process (GO:0055114); *Zm00001d005889*, oxidation-reduction process (GO:0055114); *Zm00001d006296*, regulation of translational fidelity (GO:0006450); *Zm00001d006443*, sister chromatid cohesion (GO:0007062); *Zm00001d006494*, response to karrikin (GO:0080167).

A total of 10 candidate genes were selected based on the structural variation of the gene promoter region and CDS region. There are three genes that have both genetic structure variation and promoter region variation: *Zm00001d005803*, *Zm00001d005889*, and *Zm00001d006494*. Thus, we suggest that *Zm00001d005803*, *Zm00001d005818*, *Zm00001d005823*, *Zm00001d005888*, *Zm00001d005889*, *Zm00001d006274*, *Zm00001d006296*, *Zm00001d006443*, *Zm00001d006494* and *Zm00001d006587* may be important candidate genes for qLA02-01.

## Candidate genes prediction in major QTL qLA08-01 region

We found that 29 genes were variated in the coding region, by comparing the genomic sequence differences in the qLA08-01 region between the two parents, of which 18 stop gained SNP, 1 start lost SNP were related to 14 genes. What's more, 25 InDel were detected, including 1 start lost InDel, 1 stop lost InDel and 23 frame_shift InDel, related to 18 genes. Three of those 18 genes were displayed in SNP mutation as well.

12 GO enriched genes were screened in the major qLA08-01 region (Table 5), by analysis of biological processes in the GO annotation clustering results, combined KEGG metabolic pathway analysis. Besides, based on the promoter region mutation in Table 6 and 2 genes out of 17 shown significantly variations in gene expression levels (Fig 6).

**Table 3. Selected candidate genes of leaf angle in qLA02-01 with mutations in the CDS region.**

| Gene ID | GO term | Description | Codon change | Effect |
|---|---|---|---|---|
| *Zm00001d005614* | oxidation-reduction process (GO:0055114); | Bifunctional protein FolD 1 mitochondrial | tga/ | FRAME_SHIFT |
| *Zm00001d005682* | glucuronoxylan metabolic process (GO:0010413); | Protein kinase superfamily protein | atc/ | FRAME_SHIFT |
| *Zm00001d005785* | isopentenyl diphosphate biosynthetic process, methylerythritol 4-phosphate pathway (GO:0019288); | Pentatricopeptide repeat-containing protein | att/aTtt | FRAME_SHIFT |
| *Zm00001d005779* | Biological Process: regulation of transcription, DNA-templated (GO:0006355); | ubiquitin carrier protein 7 | Gga/Tga | STOP_GAINED |
| *Zm00001d005792* | negative regulation of catalytic activity (GO:0043086); | amidase1 | gag/ | FRAME_SHIFT |
| ***Zm00001d005803*** | auxin-activated signaling pathway (GO:0009734); | SAUR-like auxin-responsive protein family | gcc/gcGGc | FRAME_SHIFT |
| *Zm00001d005808* | phosphorylation (GO:0016310); | Probable ethanolamine kinase | cgt/cgTt | FRAME_SHIFT |
| *Zm00001d005812* | sister chromatid cohesion (GO:0007062); | sterile alpha motif (SAM) domain-containing protein | tta/ttTa | FRAME_SHIFT |
| *Zm00001d005866* | cell redox homeostasis (GO:0045454); | Protein disulfide-isomerase like 2–2 | ttg/ttTTg | FRAME_SHIFT |
| ***Zm00001d005888*** | response to abscisic acid (GO:0009737); | Heat stress transcription factor B-3 | taa/ | FRAME_SHIFT |
| ***Zm00001d005889*** | oxidation-reduction process (GO:0055114); | abscisic acid 8'-hydroxylase 5 | tgccgg/aca/ | FRAME_SHIFT |
| *Zm00001d005908* | response to wounding (GO:0009611); | Tyrosine-sulfated glycopeptide receptor 1 | ttt/ttGt | FRAME_SHIFT |
| *Zm00001d005925* | starch biosynthetic process (GO:0019252); | Glucose-6-phosphate isomerase 1 chloroplastic | gtt/ | FRAME_SHIFT |
| *Zm00001d005932* | oxidation-reduction process (GO:0055114); | Aldose reductase | acg/aAcg | FRAME_SHIFT |
| *Zm00001d006027* | regulation of transcription, DNA-templated (GO:0006355); | bZIP transcription factor family protein | gct/tatgac/tcc/tccTCCGTCC | FRAME_SHIFT |
| *Zm00001d006153* | defense response by callose deposition (GO:0052542); | 1-acylglycerol-3-phosphate O-acyltransferase | cca/ | FRAME_SHIFT |
| *Zm00001d006172* | Biological Process: protein phosphorylation (GO:0006468); | Serine/threonine-protein kinase Rio1 | tGg/tAg | STOP_GAINED |
| *Zm00001d006193* | oxidation-reduction process (GO:0055114); | cytochrome P450 family 78 subfamily A polypeptide 8 | gcc/ | FRAME_SHIFT |
| ***Zm00001d006274*** | auxin-activated signaling pathway (GO:0009734); | Auxin-responsive protein SAUR61 | ggt/gAgt | FRAME_SHIFT |
| *Zm00001d006295* | protein phosphorylation (GO:0006468); | DNA-binding bromodomain-containing protein | ttg/ttgCTTGTTG | FRAME_SHIFT |
| *Zm00001d006344* | regulation of transcription, DNA-templated (GO:0006355); | Protein SUPPRESSOR OF FRI 4 | ggt/ggGt | FRAME_SHIFT |
| *Zm00001d006389* | membrane fusion (GO:0006944); | small G protein family protein / RhoGAP family protein | ttt/att/ | FRAME_SHIFT |
| *Zm00001d006437* | Biological Process: cell redox homeostasis (GO:0045454); | Monothiol glutaredoxin-S17 | atG/atA | START_LOST |
| *Zm00001d006476* | glycolytic process (GO:0006096); | aconitase5 | cct/ccTt | FRAME_SHIFT |
| ***Zm00001d006494*** | response to karrikin (GO:0080167); | Protein DETOXIFICATION 40 | gctttcctctcttttttttttccc/aag/aAGGTagatt/aTttctt/ttc/ttCCCCc | FRAME_SHIFT |
| *Zm00001d006536* | Biological Process: phosphorylation (GO:0016310); | Cysteine-rich receptor-like protein kinase 10 | Gaa/Taa | STOP_GAINED |
| *Zm00001d006548* | response to wounding (GO:0009611); | Histone H2A | ctgctt/ | FRAME_SHIFT |
| *Zm00001d006586* | response to salt stress (GO:0009651); | Peptidyl-prolyl cis-trans isomerase Pin1 | tcccgcccgcagctc/ | FRAME_SHIFT |
| ***Zm00001d006587*** | response to blue light (GO:0009637); | Chlorophyll a-b binding protein CP29.1 chloroplastic | atg/aAtg | FRAME_SHIFT |
| *Zm00001d006622* | Biological Process: electron transport chain (GO:0022900); | CYP72A57 | Cag/Tag | STOP_GAINED |

*(Continued)*

**Table 3.** (Continued)

| Gene ID | GO term | Description | Codon change | Effect |
|---------|---------|-------------|--------------|--------|
| *Zm00001d006644* | Biological Process: circadian rhythm (GO:0007623); | MAP kinase kinase kinase27 | tgA/tgG | STOP_LOST |
| *Zm00001d006675* | DNA repair (GO:0006281); | ATP-dependent DNA helicase | gtt/gTtt | FRAME_SHIFT |
| *Zm00001d006681* | negative regulation of transcription, DNA-templated (GO:0045892); | unknown | atg/ | START_LOST |

In all, 14 candidate genes were filtered as candidate gene. And the result indicates that *Zm00001d009622* (Biological Process: transcription, DNA-templated (GO:0006351)), *Zm00001d009642* (Biological Process: membrane fusion (GO:0006944)), *Zm00001d009671* (Biological Process: potassium ion transmembrane transport (GO:0071805)), *Zm00001d009676* (Biological Process: protein phosphorylation (GO:0006468)), *Zm00001d009730* (Biological Process: translation (GO:0006412)), *Zm00001d009737* (Biological Process: GTP catabolic process (GO:0006184)), *Zm00001d009754* (Biological Process: cell wall macromolecule catabolic process (GO:0016998)), *Zm00001d009789* (Biological Process: borate transport (GO:0046713)), *Zm00001d009802*(Biological Process: protein transport (GO:0015031)), *Zm00001d009871* (Biological Process: transmembrane transport (GO:0055085); Biological Process: GDP-mannose transport (GO:0015784)), *Zm00001d009948*(Biological Process: response to heat (GO:0009408)), *Zm00001d009962* (Biological Process: protein import into nucleus (GO:0006606)), *Zm00001d009610* (Acetamidase / Formamidase family protein), *Zm00001d009835* (Triosephosphate isomerase cytosolic) may be important candidate genes for qLA08-01.

**Table 4. Selected candidate genes of leaf angle in qLA02-01 with mutations in the promoter region.**

| Gene ID | GO term | Description | Effect |
|---------|---------|-------------|--------|
| *Zm00001d005682* | glucuronoxylan metabolic process (GO:0010413); | Protein kinase superfamily protein | UPSTREAM |
| ***Zm00001d005803*** | auxin-activated signaling pathway (GO:0009734); | SAUR-like auxin-responsive protein family | UPSTREAM |
| *Zm00001d005808* | phosphorylation (GO:0016310); | Probable ethanolamine kinase | UPSTREAM |
| ***Zm00001d005818*** | response to desiccation (GO:0009269); | Aldehyde dehydrogenase family 7 member B4 | UPSTREAM |
| ***Zm00001d005823*** | oxidation-reduction process (GO:0055114); | Flavonoid 3-monooxygenase | UPSTREAM |
| *Zm00001d005888* | endoplasmic reticulum unfolded protein response (GO:0030968); | Heat stress transcription factor B-3 | UPSTREAM |
| ***Zm00001d005889*** | oxidation-reduction process (GO:0055114); | abscisic acid 8'-hydroxylase5 | UPSTREAM |
| *Zm00001d006027* | endoplasmic reticulum unfolded protein response (GO:0030968); | bZIP transcription factor family protein | UPSTREAM |
| *Zm00001d006036* | response to salt stress (GO:0009651); | Heat shock 70 kDa protein 9 mitochondrial | UPSTREAM |
| *Zm00001d006153* | toxin catabolic process (GO:0009407); | 1-acylglycerol-3-phosphate O-acyltransferase | UPSTREAM |
| *Zm00001d006285* | auxin-activated signaling pathway (GO:0009734); | SAUR52-auxin-responsive SAUR family member | UPSTREAM |
| ***Zm00001d006296*** | regulation of translational fidelity (GO:0006450); | Valine—tRNA ligase mitochondrial 1 | UPSTREAM |
| *Zm00001d006389* | membrane fusion (GO:0006944); | small G protein family protein / RhoGAP family protein | UPSTREAM |
| ***Zm00001d006443*** | sister chromatid cohesion (GO:0007062); | P-loop containing nucleoside triphosphate hydrolases superfamily protein | UPSTREAM |
| *Zm00001d006467* | cysteine biosynthetic process (GO:0019344); | adenosine 5'-phosphosulfate reductase-like2 | UPSTREAM |
| ***Zm00001d006494*** | response to karrikin (GO:0080167); | Protein DETOXIFICATION 40 | UPSTREAM |
| *Zm00001d006629* | protein phosphorylation (GO:0006468); | Mitochondrial transcription termination factor family protein | UPSTREAM |
| *Zm00001d006631* | transmembrane transport (GO:0055085); | Organic cation/carnitine transporter 7 | UPSTREAM |
| *Zm00001d006646* | phosphorylation (GO:0016310); | unknown | UPSTREAM |
| *Zm00001d006688* | transmembrane transport (GO:0055085); | Putative polyol transporter 1 | UPSTREAM |
| *Zm00001d006700* | root development (GO:0048364); | mTERF family protein | UPSTREAM |

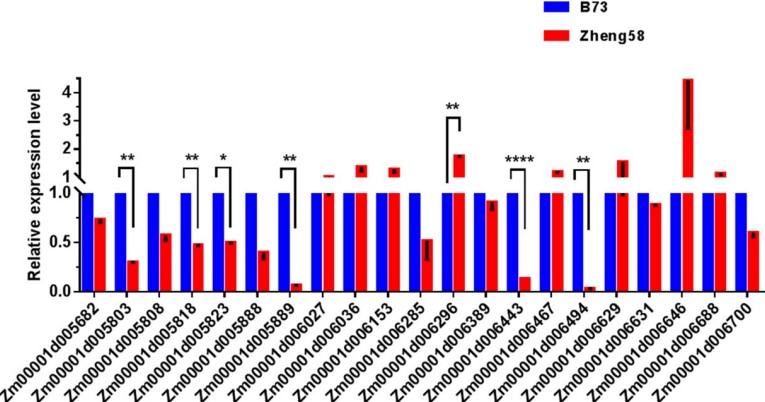

**Fig 5. The expression levels of the candidate genes on qLA02-01 loci in the two parents, B73 and Zheng58.** The expression level of each candidate gene in B73 is set to 1. The data is the mean ± SD (n = 3). P<0.05 (Student's t-test).

## Conclusion

In conclusion, the study of genetic basis of leaf angle is important for maize breeding. In the study, eight QTL for leaf angle were detected and most QTL were validated through HIF

**Table 5. Selected candidate genes of leaf angle in qLA08-01 with mutations in the CDS region.**

| Gene ID | GO term | Description | Codon change | Effect |
|---|---|---|---|---|
| *Zm00001d009622* | Biological Process: transcription, DNA-templated (GO:0006351); | Putative AP2/EREBP transcription factor superfamily protein | atc/ atcGGCGCCCGCATGACGCGGAAGCGCGgct/ tcc/tccTatg/ | FRAME_SHIFT FRAME_SHIFT FRAME_SHIFT START_LOST |
| *Zm00001d009642* | Biological Process: membrane fusion (GO:0006944); | F-box protein | cta/ | FRAME_SHIFT |
| *Zm00001d009671* | Biological Process: potassium ion transmembrane transport (GO:0071805); | Potassium transporter 10 | tGg/tAg | STOP_GAINED |
| *Zm00001d009676* | Biological Process: protein phosphorylation (GO:0006468); | serine/threonine protein kinase 3 | gac/gaCctcggac/ | FRAME_SHIFT |
| *Zm00001d009730* | Biological Process: translation (GO:0006412); | unknown | aaa/ | FRAME_SHIFT |
| *Zm00001d009737* | Biological Process: GTP catabolic process (GO:0006184); | Tubulin beta-2 chain | Cag/Tag | STOP_GAINED |
| *Zm00001d009754* | Biological Process: cell wall macromolecule catabolic process (GO:0016998); | unknown | tac/ | FRAME_SHIFT |
| *Zm00001d009789* | Biological Process: borate transport (GO:0046713); | Boron transporter 4 | agg/ | FRAME_SHIFT |
| *Zm00001d009802* | Biological Process: protein transport (GO:0015031); | Putative homeobox DNA-binding and leucine zipper domain family protein | cta/cCCta | FRAME_SHIFT |
| *Zm00001d009871* | Biological Process: transmembrane transport (GO:0055085); Biological Process: GDP-mannose transport (GO:0015784); | GDP-mannose transporter GONST1 | gga/ | FRAME_SHIFT |
| *Zm00001d009948* | Biological Process: response to heat (GO:0009408); | Heat shock 70 kDa protein 14 | cgagga/ | FRAME_SHIFT |
| *Zm00001d009962* | Biological Process: protein import into nucleus (GO:0006606); | Sas10/Utp3/C1D family | gtt/gGtt | FRAME_SHIFT |

**Table 6. Selected candidate genes of leaf angle in qLA08-01 with mutations in the promoter region.**

| Gene ID | GO term | Description | Effect |
|---|---|---|---|
| *Zm00001d009580* | | Urease accessory protein G | UPSTREAM |
| *Zm00001d009593* | | unknown | UPSTREAM |
| *Zm00001d009594* | | Aspartic proteinase A1 | UPSTREAM |
| ***Zm00001d009610*** | | Acetamidase/Formamidase family protein | UPSTREAM |
| *Zm00001d009611* | | unknown | UPSTREAM |
| *Zm00001d009612* | | DUF1645 family protein | UPSTREAM |
| *Zm00001d009619* | | Putative WRKY DNA-binding domain superfamily protein | UPSTREAM |
| *Zm00001d009620* | | Probable protein phosphatase 2C 33 | UPSTREAM |
| *Zm00001d009622* | Biological Process: transcription, DNA-templated (GO:0006351); | Putative AP2/EREBP transcription factor superfamily protein | UPSTREAM |
| *Zm00001d009631* | | CTP synthase family protein | UPSTREAM |
| *Zm00001d009679* | | ATP-dependent DNA helicase | UPSTREAM |
| *Zm00001d009704* | | | UPSTREAM |
| *Zm00001d009813* | | Nucleotide/sugar transporter family protein | UPSTREAM |
| ***Zm00001d009835*** | | Triosephosphate isomerase cytosolic | UPSTREAM |
| *Zm00001d010000* | | Thioredoxin-like 2–2 chloroplastic | UPSTREAM |
| *Zm00001d010009* | Biological Process: translation (GO:0006412); | ribosomal protein L17a | UPSTREAM |
| *Zm00001d010152* | | Histone deacetylase 8 | UPSTREAM |

approach. Candidate gene analysis within the qLA02-01 and qLA08-01 regions was conducted by whole-genome re-sequencing and expression analysis and twenty-four candidate genes for leaf angle were predicted. This study provides a better understanding of the genetic basis of leaf angle. To validate the functionality of candidate genes, future studies should be conducted using RNA-seq at the key developmental stage of maize leaf angle formation. This should alleviate inaccuracies due to differences in DNA sequences between the two parents. Furthermore, moderate fine mapping is more operable to eliminate the blindness of the authentic gene identification. Techniques such as gene editing could be utilized to edit the allele of our candidate genes, which could identify the authentic genes for qLA02-01 and qLA08-01. The cloning and function research of genes in qLA02-01 and qLA08-01 would improve our knowledge about plant architecture, and also can supply good candidate genes for molecular breeding of crops.

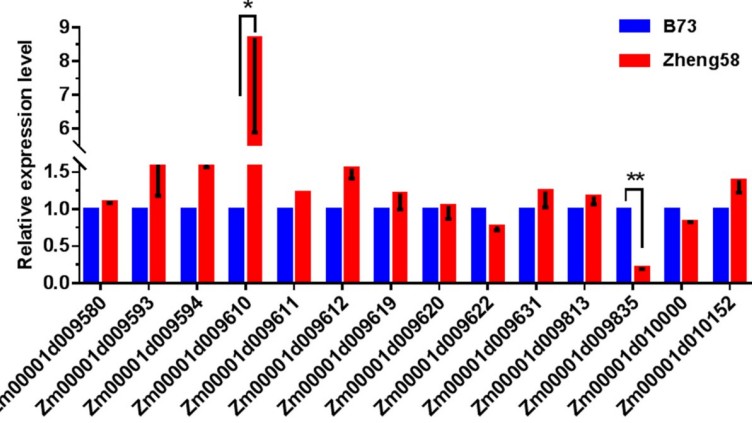

**Fig 6. The expression levels of the candidate gene on qLA08-01 loci in the two parents., B73 and Zheng58.** The expression level of each candidate gene in B73 is set to 1. The data is the mean ± SD (n = 3). P<0.05 (Student's t-test).

## Supporting information

**S1 Fig. The basic situation of the sequencing depth distribution.**
(DOCX)

**S2 Fig. Genome wide distribution of read coverage.** The horizontal axis is the chromosomal position, and the vertical axis is the median of read density of the corresponding position on the chromosome (log (2)). There is no significant difference at the 5% level. Error bars indicate the standard deviation of the phenotypic values for each genotype.
(DOCX)

**S1 Table. RT-qPCR primer.**
(DOCX)

**S2 Table. Whole genome resequencing results of Zheng58.**
(DOCX)

## Acknowledgments

The authors are grateful to Institute of Crop Science, Chinese Academy of Agricultural Sciences (ICS, CAAS) and Chinese Crop Germplasm Resources Information System (CGRIS) for providing seeds of the maize inbred lines for experiment. We thank the BioMarker Technologies Company for providing sequencing services.

## Author Contributions

**Conceptualization:** Ning Zhang.

**Formal analysis:** Ning Zhang.

**Investigation:** Ning Zhang.

**Software:** Ning Zhang.

**Writing – original draft:** Ning Zhang, Xueqing Huang.

**Writing – review & editing:** Ning Zhang, Xueqing Huang.

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
