## [Decision Letter · Decision Letter 0]

19 Oct 2020

PONE-D-20-25942

Mapping quantitative trait loci and predicting candidate genes for leaf angle in maize

PLOS ONE

Dear Dr. Huang,

Thank you for submitting your manuscript to PLOS ONE. After careful consideration, we feel that it has merit but does not fully meet PLOS ONE’s publication criteria as it currently stands. Therefore, we invite you to submit a revised version of the manuscript that addresses the points raised during the review process.

We look forward to receiving your revised manuscript.

Kind regards,

Maoteng Li

Academic Editor

PLOS ONE

Journal Requirements:

Reviewers' comments:

Reviewer's Responses to Questions

**Comments to the Author**

1. Is the manuscript technically sound, and do the data support the conclusions?

Reviewer #1: Yes

Reviewer #2: Partly

2. Has the statistical analysis been performed appropriately and rigorously? 

Reviewer #1: Yes

Reviewer #2: N/A

3. Have the authors made all data underlying the findings in their manuscript fully available?

Reviewer #1: Yes

Reviewer #2: Yes

4. Is the manuscript presented in an intelligible fashion and written in standard English?

Reviewer #1: Yes

Reviewer #2: Yes

5. Review Comments to the Author

Reviewer #1: Please see more comments in the attached PDF. Those marked handwriting comments and suggestions should be addressed to improve the manuscripts. Some figures should be edited and reupload for publication if accepted.

Reviewer #2: The manuscript mapped 8 major QTLs with F3:4 recombinant inbred lines (RIL) population derived from B73 x Zheng58 for leaf angle and confirmed some of them through a heterogeneous inbred family (HIF) approach. Candidate genes for the qLA02-01 and qLA08-01 regions, with largest contribution effect, were predicted through bioinformatics analysis. The results will benefit to clone the favorable allele for leaf angle or develop the novel maize varieties with ideal plant architecture through marker-assisted selection.

Major issues

Q1 some basic mistakes

L18

Does "biomass" mean “harvest index" here? Because high biomass may not guarantee high grain yield in maize and, in most cases, I believe, it is the grain yield, not biomass, is the major goal for corn production.

L22

Does the leaf angle refer in particular to that of " the upper leaves of the ear"？

L30

11 is the number of all genes that affect leaf angle in maize? or this is only an incomplete statistical number? In my opinion, there are far more known genes involved in leaf angel control in maize. such as CT2 (https://doi. org/10.1371/journal.pgen.1007374)

L173

I guess the auther want to say the B73 allels conribute to the expanded leaf angel with "additive effect". However, it is easiy to lead confusion to change the meaning of a commonly used concept .

Q2

L71

Since phenotypes of quantitative traits are easily affected by environment factors, details of the condition during growing seasons, including coordinate, ptotoperiod, temperature and so on should be provided.

Q3

L69

Were the 165 lines used in the experiment from one F2 ear or 165 F2 ears?

Q4

L75,76

RIL population with 165 lines is small. You may lose some lines during growing in field experiment. Furthermore, phenotypes of quantitative traits are easily affected by QTL X environment effects. However, only two replications were carried out. I need to know how many lines showed good repeatability ? That directly determines the reliability of the data.

Q5

L230,L272

When do the candidate genes prediction, you may eliminate more irrelevant genes if you take the synonymous mutation into account when select candidate genes in CDS regions.

Q6

Are there any cloned genes that control the leaf angle locate in QTL regions mapped in the manuscript?

Q7

L308

“techniques such as gene editing could be utilized to edit the allele of our candidate genes, which could identify the authentic genes for qLA02-01 and qLA083-1.”

Gene editing do can be utilized to edit the allele of candidate genes. However, in this situation, I believe moderate fine mapping is more operable to eliminate the blindness of the authentic gene identification.

Q8

L83-85

“Three consecutive leaves were measured for each plant, including the first leaf above the primary ear, the primary ear leaf and the first leaf below the primary ear.”

which one of the three leaves was used for data collection of leaf angle? Why?

Q9

“It was noteworthy that all QTL had positive additive effects, suggesting that the B73 parent contributed most alleles for increasing leaf angle (Table 2).”

Why QTL X environment effect was not detected? It is the character of the leaf angle trait, or it resulted from the experiment design because the phenotype was surveyed in the same field, or resulted from the genetic characteristics of selected parents, zheng58 and B73.

Minor issues

Although, in general, the manuscript was written clearly enough, it still requires extensive editing. eg., There are some grammar mistakes or spelling errors in the manuscript (underlined by red lines), eg. L55, 129,145,260.

Many sentences are too wordy and need to be re-organized.

Some figures and tables, such as Fig.5,6, Table 3, are not essentially to contribute to the research results directly could be provided as supplementary data.

L320,

"The cloning and function research of genes in qLA02-01 and qLA08-01" should be more accureate than “The cloning and function research of qLA02-01 and qLA08-01”

6. PLOS authors have the option to publish the peer review history of their article (what does this mean?). If published, this will include your full peer review and any attached files.

Reviewer #1: **Yes: **Shuyu Liu

Reviewer #2: **Yes: **Yong Shi

---

## [Author Response · Author response to Decision Letter 0]

18 Nov 2020

Reviewer #1: Please see more comments in the attached PDF. Those marked handwriting comments and suggestions should be addressed to improve the manuscripts. Some figures should be edited and re-upload for publication if accepted.

Response: Thanks for your comments. According to your comments and suggestions, we have made some modifications to improve the manuscripts. Some figures have been re-edited and re-upload for publication. Please see the marked-up copy of our manuscript that highlights changes made to the original version. As the manuscript is converted to PDF format, some figures may not be displayed clearly in PDF format. However, each figure in TIFF format in submission system is of high resolution. If the manuscript is accepted, we will provide high-resolution TIFF images to meet the publishing requirements.

Q1 Line 37/76/77 should you change all numbers as letters if they are <10?

Response: Thanks for your comments. We have changed all numbers as letters if they are <10. Please see the marked-up copy of our manuscript that highlights changes made to the original version. 

Q2 Ten days after pollination (DAP)

Response: Thanks for your comments. We have inserted a space after “pollination”. Please see Line 86 in the marked-up copy of our manuscript that highlights changes made to the original version. 

Q3 Line85 you only measured 9 data points per line, right? Is it too few?

Response: Thanks for your comments. In our work three leaves were measured of each plant. Considering maize plant architecture can be typically evaluated by leaf angle of the three leaves: the first leaf above the primary ear, the primary ear leaf and the first leaf below the primary ear. We randomly choose three out of five plant of each line used for collecting leaf angle data, which can represent the phenotypic value of leaf angle for each line. 

Q4 line 151 table1 h2/%?

Response: Thanks for your comments. We have deleted “/%” and inserted “(%)”. Please see line 158 table1 in the marked-up copy of our manuscript that highlights changes made to the original version. 

Q4 line159-163 Fig 2 this is not a high resolution map. Resolution of Fig 2 is very low, need to redo.

Response: Thanks for your comments. We have re-upload a high resolution map for Fig 2. 

Q5 Line 183 also the number of SSRs used for mapping? Line183-185 better mention corresponding mbp.

Response: Thanks for your comments. In our work, all of the 189 SSR markers with polymorphic between the two parents were used for mapping. In Line183-185, the corresponding Mbp were supplemented. Please see Line191-195 in the marked-up copy of our manuscript that highlights changes made to the original version.

Q6 line 190 Fig 3 use chr or chrom instead of chro for abbreviation.

Response: Thanks for your comments. We used chr instead of chro for abbreviation. Please see Fig 3. 

Q7 line 200 Theoretically, the leaf angle trait in progenies from selfing HIF155 will be only segregated for qLA01-01.

Response: Thanks for your comments. We have changed “selfing” into “selfed”. Please see Line 209 in the marked-up copy of our manuscript that highlights changes made to the original version. 

Q8 line 233 explain more?

Response: Thanks for your comments. Absolutely, stop gained and start lost, as well as frame-shift mutations often have a greater impact on causing changes in gene function. Thence, these mutations stimulated our further research interest. Please see Line 246-248 in the marked-up copy of our manuscript that highlights changes made to the original version. 

Q9 ling316-322 this should be in discussion.

Response: Thanks for your comments. These sentences mainly explained the future studies for the qLA02-01 and qLA08-01. There is no particular “Discussion” section in the main text, so we think it's appropriate to put them here.

Reviewer #2: The manuscript mapped 8 major QTLs with F3:4 recombinant inbred lines (RIL) population derived from B73 x Zheng58 for leaf angle and confirmed some of them through a heterogeneous inbred family (HIF) approach. Candidate genes for the qLA02-01 and qLA08-01 regions, with largest contribution effect, were predicted through bioinformatics analysis. The results will benefit to clone the favorable allele for leaf angle or develop the novel maize varieties with ideal plant architecture through marker-assisted selection.

Major issues

Q1 some basic mistakes

L18

Does "biomass" mean “harvest index" here? Because high biomass may not guarantee high grain yield in maize and, in most cases, I believe, it is the grain yield, not biomass, is the major goal for corn production.

Response: Thanks for your comments. The grain yield, not biomass, is the major goal for corn production. We deleted "biomass". Please see Line 20 in the marked-up copy of our manuscript that highlights changes made to the original version. 

L22

Does the leaf angle refer in particular to that of " the upper leaves of the ear"？

Response: Thanks for your comments. In this sentence, the leaf angle refer in particular to the upper leaves of the ear.

L30

11 is the number of all genes that affect leaf angle in maize? or this is only an incomplete statistical number? In my opinion, there are far more known genes involved in leaf angel control in maize. such as CT2 (https://doi. org/10.1371/journal.pgen.1007374)

Response: Thanks for your comments. Just as you mentioned, such as CT2, some other genes also affect leaf angle. Here we only listed 11 genes because they were representative genes identified by QTL by mutagenesis and QTL-cloning approach. We modified the sentence. Please see Line 32-33 in the marked-up copy of our manuscript that highlights changes made to the original version.

L173

I guess the auther want to say the B73 allels conribute to the expanded leaf angel with "additive effect". However, it is easiy to lead confusion to change the meaning of a commonly used concept .

Response: Thanks for your comments. Here, "additive effect" is easily to lead confusion. We changed "additive effect" to "additive allele effect". Please see Line 180 and Table 2 in the marked-up copy of our manuscript that highlights changes made to the original version.

Q2

L71

Since phenotypes of quantitative traits are easily affected by environment factors, details of the condition during growing seasons, including coordinate, ptotoperiod, temperature and so on should be provided.

Response: Thanks for your comments. Environment factors can affect the leaf angle. We supplemented the planting and growing season in the section “Field experiments and statistical analyses”. Please see the Line 78-79 in the marked-up copy of our manuscript that highlights changes made to the original version.

Q3

L69

Were the 165 lines used in the experiment from one F2 ear or 165 F2 ears?

Response: Thanks for your comments. The165 lines were selected from one F2 ear. 

Q4

L75,76

RIL population with 165 lines is small. You may lose some lines during growing in field experiment. Furthermore, phenotypes of quantitative traits are easily affected by QTL X environment effects. However, only two replications were carried out. I need to know how many lines showed good repeatability ? That directly determines the reliability of the data.

Response: Thanks for your comments. In this work, more than two hundred lines were planted and 165 lines were survival. Considering that the phenotype of leaf angle was easily affected by environmental conditions, we planted these 165 lines in the same season and the same field in 2014 and 2015. So, the environmental factors affecting leaf angle phenotype were minimized. These 165 lines showed good repeatability in the phenotype of leaf angle and the phenotypic data were used to detected the QTL for leaf angle. Subsequently, we construct the HIF populations to verify the major QTL, each HIF population with at least 120 plants. These plants were also grown in the same season and the same field in 2016. The results verified the authenticity of the QTL mapping results and showed the reliability of the data.

Q5

L230,L272

When do the candidate genes prediction, you may eliminate more irrelevant genes if you take the synonymous mutation into account when select candidate genes in CDS regions.

Response: Thanks for your comments. Yes, we eliminated many irrelevant genes by taking the synonymous mutation into account when select candidate genes in CDS. Please see the Line 241and Line285 in the marked-up copy of our manuscript that highlights changes made to the original version.

Q6

Are there any cloned genes that control the leaf angle locate in QTL regions mapped in the manuscript?

Response: Thanks for your comments. Yes. Just as shown in Fig 3, some genes has been cloned that control the leaf angle located in the QTL we detected, such as UPA1, lg1 and so on. 

Q7

L308

“techniques such as gene editing could be utilized to edit the allele of our candidate genes, which could identify the authentic genes for qLA02-01 and qLA083-1.”

Gene editing do can be utilized to edit the allele of candidate genes. However, in this situation, I believe moderate fine mapping is more operable to eliminate the blindness of the authentic gene identification.

Response: Thanks for your comments. We supplemented your suggestion in the text. Please see the line 331 in the marked-up copy of our manuscript that highlights changes made to the original version.

Q8

L83-85

“Three consecutive leaves were measured for each plant, including the first leaf above the primary ear, the primary ear leaf and the first leaf below the primary ear.”

which one of the three leaves was used for data collection of leaf angle? Why?

Response: Thanks for your comments. In this work, we measured three leaves and used the mean value for data analysis. Considering maize plant architecture can be typically evaluated by leaf angle of the three leaves: the first leaf above the primary ear, the primary ear leaf and the first leaf below the primary ear. We randomly choose three out of five plant of each line used for collecting leaf angle data. 

Q9

“It was noteworthy that all QTL had positive additive effects, suggesting that the B73 parent contributed most alleles for increasing leaf angle (Table 2).”

Why QTL X environment effect was not detected? It is the character of the leaf angle trait, or it resulted from the experiment design because the phenotype was surveyed in the same field, or resulted from the genetic characteristics of selected parents, zheng58 and B73.

Response: Thanks for your comments. Considering that the phenotype of leaf angle was easily affected by environmental conditions, we planted these 165 lines in the same season and the same field in 2014 and 2015. These 165 lines showed good repeatability in the phenotype of leaf angle and the phenotypic data were used to detected the QTL for leaf angle. The environmental factors affecting leaf angle phenotype were minimized and the QTL X environment effect was not detected. Subsequently, we construct the HIF populations to verify the major QTL, each HIF population with at least 120 plants. These plants were also grown in the same season and the same field in 2016. The results verified the authenticity of the QTL mapping results and showed that these major QTL are stable and inheritable. In this work, the character of the leaf angle trait resulted from the genetic characteristics of selected parents, zheng58 and B73.

Minor issues

Although, in general, the manuscript was written clearly enough, it still requires extensive editing. eg., There are some grammar mistakes or spelling errors in the manuscript (underlined by red lines), eg. L55, 129,145,260.

Response: Thanks for your comments. We have made some modifications for grammar and spelling. Please see the marked-up copy of our manuscript that highlights changes made to the original version. 

Many sentences are too wordy and need to be re-organized.

Some figures and tables, such as Fig.5,6, Table 3, are not essentially to contribute to the research results directly could be provided as supplementary data.

Response: Thanks for your comments. We have made some modifications for many sentences. Fig.5,6 and Table 3 were removed to supplementary data. Please see S1 Fig., S2 Fig. and S2 Table. in the marked-up copy of our manuscript that highlights changes made to the original version. 

L320,

"The cloning and function research of genes in qLA02-01 and qLA08-01" should be more accureate than “The cloning and function research of qLA02-01 and qLA08-01”

Response: Thanks for your comments. We made the modification according to your suggestion. Please see Line 333 in the marked-up copy of our manuscript that highlights changes made to the original version.

---

## [Decision Letter · Decision Letter 1]

14 Dec 2020

PONE-D-20-25942R1

Mapping quantitative trait loci and predicting candidate genes for leaf angle in maize

PLOS ONE

Dear Dr. Huang,

Thank you for submitting your manuscript to PLOS ONE. After careful consideration, we feel that it has merit but does not fully meet PLOS ONE’s publication criteria as it currently stands. Therefore, we invite you to submit a revised version of the manuscript that addresses the points raised during the review process.

We look forward to receiving your revised manuscript.

Kind regards,

Maoteng Li

Academic Editor

PLOS ONE

Reviewers' comments:

Reviewer's Responses to Questions

**Comments to the Author**

1. If the authors have adequately addressed your comments raised in a previous round of review and you feel that this manuscript is now acceptable for publication, you may indicate that here to bypass the “Comments to the Author” section, enter your conflict of interest statement in the “Confidential to Editor” section, and submit your "Accept" recommendation.

Reviewer #1: All comments have been addressed

Reviewer #2: (No Response)

2. Is the manuscript technically sound, and do the data support the conclusions?

Reviewer #1: Yes

Reviewer #2: Yes

3. Has the statistical analysis been performed appropriately and rigorously? 

Reviewer #1: Yes

Reviewer #2: Yes

4. Have the authors made all data underlying the findings in their manuscript fully available?

Reviewer #1: Yes

Reviewer #2: Yes

5. Is the manuscript presented in an intelligible fashion and written in standard English?

Reviewer #1: Yes

Reviewer #2: Yes

6. Review Comments to the Author

Reviewer #1: (No Response)

Reviewer #2: L22 “The less of leaf angle is, the more upright the upper leaves of the ear are.” I believe “The less of leaf angle is, the more upright the leaves are.” is more accurate. The author is descripting a general concept here, and could not explain it refer in particular to a situation. In case of a particular situation, you may explain it separately. For example, you may say’ The less of leaf angle is, the more upright the leaves are. Upright upper leaves of the ear can maximize photosynthesis efficiency…’here.

L173

Actually, Additive allele effect and Additive effect are with the same meaning to me. The author is still descripting a general concept refer in particular to a situation. To avoid confusion, you may not need to explain Additive effect here, “A positive additive effect value indicates that the B73 allele increases the value of the trait; A negative value indicates that the Zheng58 allele increases the value of the trait.” is clearly enough.

Q3

L69

I do not understand. If the165 lines were selected from one F2 ear (kernels in the ear are F3 generation), then the single seed descents were from the F3, not F2. Usually, F2 population is a good representative of extensive separation for genes of two parents.

Q4

L75 76

It is better to put the sentence in the manuscript, especially “more than two hundred lines were planted and 165 lines were survival.” to indicate the reliability of data

Q8

L83-85

Put the sentence” we measured three leaves and used the mean value for data analysis.” in the manuscript in order other researchers could repeat the trial.

Q9

Although I still believe there should be some QTL X environment effect even the environmental conditions were controlled, if it is not the character of the leaf angle trait, because environmental deviation should exist between years. However, the explication by the author seems acceptable, so put the explication in the manuscript.

7. PLOS authors have the option to publish the peer review history of their article (what does this mean?). If published, this will include your full peer review and any attached files.

Reviewer #1: **Yes: **Shuyu Liu

Reviewer #2: **Yes: **Yong Shi

---

## [Author Response · Author response to Decision Letter 1]

15 Dec 2020

Reviewer #2: 

Q1 

L22

 “The less of leaf angle is, the more upright the upper leaves of the ear are.” I believe “The less of leaf angle is, the more upright the leaves are.” is more accurate. The author is descripting a general concept here, and could not explain it refer in particular to a situation. In case of a particular situation, you may explain it separately. For example, you may say’ The less of leaf angle is, the more upright the leaves are. Upright upper leaves of the ear can maximize photosynthesis efficiency…’here.

Response: Thanks for your comments. We have delete “The less of leaf angle is, the more upright the upper leaves of the ear are.” And we have inserted “The less of leaf angle is, the more upright the leaves are.” Please see Line 21 and line 22 in the marked-up copy of our manuscript that highlights changes made to the original version. 

L173

Actually, Additive allele effect and Additive effect are with the same meaning to me. The author is still descripting a general concept refer in particular to a situation. To avoid confusion, you may not need to explain Additive effect here, “A positive additive effect value indicates that the B73 allele increases the value of the trait; A negative value indicates that the Zheng58 allele increases the value of the trait.” is clearly enough.

Response: Thanks for your comments. Under the table we have deleted “effect of the situation of the Zheng58 allele by the B73 allele”. And just use “A positive additive effect value indicates that the B73 allele increases the value of the trait; A negative value indicates that the Zheng58 allele increases the value of the trait.” as the explanation. Please see the Line178 in the marked-up copy of our manuscript that highlights changes made to the original version.

Q3

L69

I do not understand. If the165 lines were selected from one F2 ear (kernels in the ear are F3 generation), then the single seed descents were from the F3, not F2. Usually, F2 population is a good representative of extensive separation for genes of two parents.

Response: Thanks for your comments. Here we did not describe accurately. We have reorganized our description just as “A single seed descent from one F1 progeny and then two generations of self-pollination were applied to produce the recombination inbred line population with 165 lines.” Please see the Line69-70 in the marked-up copy of our manuscript that highlights changes made to the original version.

Q4

L75 76

It is better to put the sentence in the manuscript, especially “more than two hundred lines were planted and 165 lines were survival.” to indicate the reliability of data

Response: Thanks for your comments. We have inserted the sentence “more than two hundred lines were planted and 165 lines were survival.” Please see Line76-77 in the marked-up copy of our manuscript that highlights changes made to the original version. 

Q8

L83-85

Put the sentence” we measured three leaves and used the mean value for data analysis.” in the manuscript in order other researchers could repeat the trial.

Response: Thanks for your comments. We have put the sentence “We measured three leaves and used the mean value for data analysis.” Please see the line 84-85 in the marked-up copy of our manuscript that highlights changes made to the original version. 

Q9

Although I still believe there should be some QTL X environment effect even the environmental conditions were controlled, if it is not the character of the leaf angle trait, because environmental deviation should exist between years. However, the explication by the author seems acceptable, so put the explication in the manuscript.

Response: Thanks for your comments. We have insert the explication into the manuscript. Please see the line218-228 in the marked-up copy of our manuscript that highlights changes made to the original version.

---

## [Editor Report · Decision Letter 2]

23 Dec 2020

Mapping quantitative trait loci and predicting candidate genes for leaf angle in maize

PONE-D-20-25942R2

Dear Dr. Huang,

We’re pleased to inform you that your manuscript has been judged scientifically suitable for publication and will be formally accepted for publication once it meets all outstanding technical requirements.

Kind regards,

Maoteng Li

Academic Editor

PLOS ONE
---

## [Editor Report · Acceptance letter]

28 Dec 2020

PONE-D-20-25942R2 

Mapping quantitative trait loci and predicting candidate genes for leaf angle in maize 

Dear Dr. Huang:

I'm pleased to inform you that your manuscript has been deemed suitable for publication in PLOS ONE. Congratulations! Your manuscript is now with our production department. 

Kind regards, 

on behalf of

Dr. Maoteng Li 

Academic Editor

PLOS ONE